# An Overview of Cannabidiol as a Multifunctional Drug: Pharmacokinetics and Cellular Effects

**DOI:** 10.3390/molecules29020473

**Published:** 2024-01-18

**Authors:** Nadia Martinez Naya, Jazmin Kelly, Giuliana Corna, Michele Golino, Ariel H. Polizio, Antonio Abbate, Stefano Toldo, Eleonora Mezzaroma

**Affiliations:** 1Robert M. Berne Cardiovascular Research Center, Division of Cardiovascular Medicine, School of Medicine, University of Virginia, Charlottesville, VA 22903, USA; zfw3js@virginia.edu (N.M.N.); jas5rc@virginia.edu (J.K.); ap9dr@virginia.edu (A.H.P.); tyz2qs@uvahealth.org (A.A.); yqd5uq@virginia.edu (S.T.); 2Interventional Cardiology Department, Hospital Italiano de Buenos Aires, Buenos Aires 1199, Argentina; giuliana.corna@hospitalitaliano.org.ar; 3Pauley Heart Center, Virginia Commonwealth University, Richmond, VA 23220, USA; michele.golino@vcuhealth.org; 4Department of Medicine and Surgery, University of Insubria, 2110 Varese, Italy; 5School of Pharmacy, Virginia Commonwealth University, Richmond, VA 23220, USA

**Keywords:** cannabidiol, CBD, pharmacokinetics, antioxidant, anti-inflammatory, cellular effects

## Abstract

Cannabidiol (CBD), a non-psychoactive compound derived from Cannabis Sativa, has garnered increasing attention for its diverse therapeutic potential. This comprehensive review delves into the complex pharmacokinetics of CBD, including factors such as bioavailability, distribution, safety profile, and dosage recommendations, which contribute to the compound’s pharmacological profile. CBD’s role as a pharmacological inhibitor is explored, encompassing interactions with the endocannabinoid system and ion channels. The compound’s anti-inflammatory effects, influencing the Interferon-beta and NF-κB, position it as a versatile candidate for immune system regulation and interventions in inflammatory processes. The historical context of Cannabis Sativa’s use for recreational and medicinal purposes adds depth to the discussion, emphasizing CBD’s emergence as a pivotal phytocannabinoid. As research continues, CBD’s integration into clinical practice holds promise for revolutionizing treatment approaches and enhancing patient outcomes. The evolution in CBD research encourages ongoing exploration, offering the prospect of unlocking new therapeutic utility.

## 1. Introduction

*Cannabis sativa*, also called Marijuana or hemp, is a plant that has been primarily used for recreational and medicinal purposes for centuries. It synthesizes numerous biological compounds; at least 554 have been described [1,2,3], including 120 terpenes, which are responsible for its characteristic aroma [3], and 113 phytocannabinoids [1,2], which are substances that possess a typical C21 terpenophenolic skeleton [2].

Cannabis remained a common ingredient in medicines during the 19th century. It was widely used in the United States and Europe, with various formulations available over the counter, until in 1937, the passage of the ‘Marihuana Tax Act’ and newly imposed penalties caused research on Cannabis to be restricted [4]. In the early 20th century, attitudes toward cannabis began to change, influenced by social and political factors. In recent decades, there has been a renewed interest in the therapeutic potential of cannabis, including its individual compounds. The primary psychoactive compound found in Cannabis is Δ*9-tetrahydrocannabidiol* (Δ9-THC) [2,5], while Cannabidiol (CBD) is the main non-psychoactive component. CBD is a small lipophilic molecule that was initially discovered in 1940 as a derivative of the cannabidiolic acid (2-[(1R,6R)-3-methyl-6-prop-1-en-2-ylcyclohex-2-en-1-yl]-5-pentylbenzene-1,3-diol).

This compound has garnered considerable attention in the biomedical field due to its reported analgesic and anti-inflammatory effects, with the added advantage of not inducing the undesired psychotropic effects of Δ9-THC. Over the past few decades, there has been a growing body of research dedicated to CBD, revealing its significant therapeutic potential. This review is centered on CBD as a phytocannabinoid, digging into its pharmacokinetics, formulations, medicinal properties, and reported cellular and molecular effects.

### The Endocannabinoid System: Cannabinoid Receptors and Endocannabinoids

The Endocannabinoid system comprises the cannabinoid G-protein-coupled receptors 1 and 2 (CB1 and CB2, respectively) and their endogenous agonists, defined as endocannabinoids [2], which are lipidic molecules. The principal endocannabinoids are arachidonoyl ethanolamide (Anandamide), 3,2-arachidonoyl glycerol, and 4–6 and 2-arachidonyl glyceryl ether (Noladin ether) [6,7]. CB1, initially isolated from rat cerebral cortex cDNA and identified as a mediator of the pharmacological effects of Δ9-THC [8,9,10], is predominantly expressed in the central nervous system, whereas CB2 has been identified for its homology to CB1 and its affinity to Δ9-THC and is mainly expressed in hematopoietic cells and peripheral tissues [11,12,13,14,15,16,17,18,19].

CBD displays multiple effects on the endocannabinoid system that are still not fully understood. CBD possesses a classical cannabinoid structure [20]. Nevertheless, it binds with low affinity to CB1 and CB2 [10,20,21,22].

Notably, certain in vivo effects of CBD, including its capacity to enhance adult neurogenesis, depend on the presence of the CB1 receptor, a conclusion supported by the lack of discernible impact in animals devoid of CB1 receptors [23]. CBD can behave as a CB2 receptor inverse agonist [24] but also exhibits a non-competitive antagonist of CB1 and CB2 in the mouse vas deferens [17,24,25]. Moreover, CBD has the potential to influence endocannabinoid signaling by binding to an allosteric site on CB1 receptors, a site with a distinct function from the orthosteric site essential for endocannabinoids like 2-Arachidonoylglycerol (2-AG), leading to a decrease in the effectiveness and strength of binding for these natural compounds [26].

## 2. CBD: Structure, Sources, and Clinical Use

Isolated from the cannabis plant in the late 1930s and early 1940s, with its structure elucidated in 1963 [27], CBD is a cannabinoid, a term commonly used to refer to chemical substances isolated from the cannabis plant [2].

CBD contains twenty-one carbon atoms, sharing the exact chemical formula with THC, C21H30O2, and a molecular weight of 314.464 g/mol [27,28], arranged into a cyclohexene ring, a phenolic ring, and a pentyl side chain. In addition, the terpenic and phenolic rings are located in planes almost perpendicular to each other [28].

Structurally, there is a critical difference between CBD and Δ9-THC: whereas the latter contains a cyclic ring, CBD includes a hydroxyl group (Figure 1). Noteworthy, the saturated exocyclic C−C double bond prevents the conversion of CBD to the psychoactive Δ9-THC. The subtle variance in the molecular structure of these two compounds results in a significant influence on their pharmacological characteristics [1,2].

CBD has potential antioxidant properties because of its cationic-free radicals, derived from its scavenging free radical reaction, and exhibits several stable resonance structures. Due to their antioxidant reaction, the unpaired electrons in the CBD molecule are distributed mainly on ether and alkyl moieties and the benzene ring (Figure 2). Thus, this resonance stabilizes the CBD-free radical, diminishing its reactivity and converting it into a long-lived and less reactive free radical species than the original radical substrate [28].

In the United States, the Food and Drug Administration (FDA) has granted approval for the use of CBD in rare seizure disorders, like Dravet and Lennox–Gastaut syndromes, along with tuberous sclerosis complex, a genetic disease leading to the growth of benign tumors in the brain and other organs [33]. In 2018, the U.S. Congress enacted the Agriculture Improvement Act, signed into law, which removed hemp from the federal Controlled Substances Act and conferred legal status upon CBD derived from hemp [34]. A 2021 trial led by Maguire et al. focused on in vitro assessments of purified natural and synthetic CBD from diverse sources; the results revealed remarkably similar effects across all experiments, highlighting the potential presence at moderate concentrations of other, potentially psychoactive phytocannabinoids in natural CBD formulations [35]. Nevertheless, the ongoing discussion regarding the effectiveness of plant-derived CBD formulations versus their synthetic equivalents remains inconclusive.

## 3. CBD as a Drug: Pharmacokinetics and Current Formulations

Numerous studies have explored the pharmacokinetics of CBD-THC combinations, but there is a scarcity of research explicitly investigating the pharmacokinetics of CBD in isolation [36]. It is noteworthy to mention that the formulation, route of administration, dosage schedule (single or multiple doses), type of formulation system, and dietary factors all play significant roles in influencing the pharmacokinetics and effects associated with cannabis-derived medications [37].

### 3.1. Absorption

Oral administration

While controlled human studies predominantly employ oral administration [36,37], CBD exhibits limited oral bioavailability, estimated to be as low as 6%, owing to its highly lipophilic nature and extensive first-pass metabolism [1,2].

Vast efforts have been made to improve the oral bioavailability of cannabinoids in patients [37], with advanced formulations like lipid/oil-based formulations, gelatin matrix pellets, and self-emulsifying drug delivery systems (SEDDS) shown to be efficient methods to increase its effectiveness [37,38].

SEDDS-CBD demonstrated a maximum plasma concentration (Cmax) up to 4.4 times higher than a commercial formulation [37]. Likewise, Patrician et al. observed increased Cmax with no differences in time to maximal absorption using a modified oral CBD formulation, incorporating DehydraTECHTM delivery technology, which involves associating long-chain fatty acids, high in oleic acid through a patented dehydration process with CBD, effectively reducing first-pass liver metabolism [39].

The time to maximum concentration (Tmax) of CBD varies from 1 to 6.13 h post-ingestion, with a half-life (t½) estimated between 18 and 32 h based on limited studies [36,40]. CBD exhibits high absorption variability, influenced by food/fat intake, enhancing its area under the curve (AUC) and maximum concentration (Cmax) [36,37,41]. Gender differences show greater absorption in women than men [42], likely due to distribution volume, body fat percentage, or hormonal variations, challenging a one-size-fits-all dosage strategy [37,42,43]. Moreover, patients with moderate or severe hepatic impairment exhibit 2.5 to 5.2-fold higher CBD AUC compared to those with normal hepatic function [44].

Inhalation

Inhalation achieves rapid peak plasma concentrations (within 3–10 min) with higher maximum levels compared to oral ingestion [45], resulting in bioavailability ranging from 11% to 45% (mean 31%) [40]. Delivery through inhalation or oro-mucosal routes minimizes or mitigates the extensive first-pass metabolism associated with oral cannabinoid administration. Using a vaporizer to administer cannabinoid compounds avoids the respiratory risks associated with smoking cannabis and the exposure to toxic pyrolytic compounds formed via combustion.

Intravenous administration

The highest plasma concentrations of CBD were obtained by intravenous administration [46]. Following intravenous administration of 20 mg of deuterium-labeled CBD, the mean plasma concentration reported was 686 ng/mL (3 min post-administration), which dropped to 48 ng/mL at one h [46,47].

Transdermal administration

CBD’s hydrophobic nature limits its diffusion across the skin’s aqueous layer. Different strategies, like chemical penetration enhancers, microemulsions, and physical enhancers, are used to improve CBD permeability [45]. This route of administration avoids the first-pass metabolism, leading to higher bioavailability rates in the presence of enhancers and prolonged steady plasma concentration compared to other routes of delivery [48,49]. There are limited published studies on transdermal cannabinoid delivery systems, but CBD gel and topical creams have been demonstrated to be successful in animal studies [50].

### 3.2. Distribution

Due to the highly lipophilic nature of cannabinoids, including CBD, they are sequestered in fatty tissues and penetrate highly vascularized tissue (such as adipose, heart, brain, liver, lungs, and spleen) with subsequent equilibration into less vascularized tissue and rapid decrease in plasma concentration [45,51]. The volume of distribution (Vd) in adults, assuming a 70 kg body weight, is moderate to high since it varies from 2.5–10 L/kg. CBD, like THC, has a protein binding capability of >95%, mainly to lipoproteins. Only 1–5% of the total concentration of CBD is unbound and has pharmacological effects [51].

### 3.3. Metabolism

CBD undergoes hepatic metabolism primarily by isozymes CYP2C19 and CYP3A4, with additional involvement of CYP1A1, CYP1A2, CYP2C9, and CYP2D6. The active metabolite, 7-hydroxy cannabidiol (7-OH-CBD), exhibits a 38% lower plasma AUC than the parent drug. Following hydroxylation, further hepatic metabolism occurs, leading to subsequent fecal and, to a lesser extent, urinary excretion of these metabolites [45].

Nevertheless, limited information is available regarding the pharmacological activity of CBD metabolites in humans [45].

### 3.4. Excretion

CBD has a prolonged terminal elimination half-life. In fact, after twice-daily dosing for seven days in healthy volunteers, the terminal half-life of CBD in plasma was 56 to 61 h [44]. The average half-life following intravenous dosing is observed to be 24 ± 6 h and post-inhalation to be 31 ± 4 h. Most CBD is excreted unchanged through feces but also as both unchanged and glucuronidated CBD in the urine [51].

### 3.5. Drug Interaction

CBD has the potential to cause interaction effects with many over-the-counter and prescription medications since it acts on cytochrome P450 isoforms (enzymes involved in the metabolism of many drugs) [36]. By inhibiting CYP3A4 and CYP2D6, CBD may lead to an increased concentration of other drugs via enhanced metabolism, which thus exaggerates the drugs’ effects and may result in substantial adverse reactions [40].

### 3.6. Safety and Adverse Effects

CBD is considered safer than other cannabinoids since it lacks cognitive and psychotropic effects [27]. Orally administered CBD is well-tolerated [52]. A comprehensive review across all indications found few mild/moderate adverse events associated with CBD, strongly linked to dosage [53]. Common side effects (≥10% incidence) include drowsiness, sedation (more frequent initially, decreasing over time), sleeping problems, fatigue, dizziness, headache, nausea, diarrhea, decreased appetite, abdominal discomfort, pain, vomiting, suicidal thoughts, changes in mood or behavior, weight loss, and rash. Severe adverse events are rare (3–10% incidence), including fever, upper respiratory tract infections, seizures, and elevated liver enzymes (ALT/AST) [54]. The liver impact depends on CBD dosage and initial transaminase levels, typically occurring in the first months of treatment, often requiring adjustment or discontinuation [33]. This is attributed to CBD impeding the hepatic metabolism of other drugs [53]. In one randomized trial [55], CBD treatment was linked to anemia (~30%) and increased serum creatinine (~10%) through unknown mechanisms [33]. Lastly, CBD can rarely cause hypersensitivity reactions treated with corticosteroids and/or antihistamines [33].

### 3.7. Dosage

CBD is available in many formulations (e.g., purified CBD, CBD: THC ratios, CBD enriched products) and forms, like oil solution, sublingual tablets, capsules, tablets, sublingual spray, nasal spray, and creams, and can be found in dietary supplements, cosmetics, and animal health products [56], with Epidiolex^®^ being the first and only FDA-approved CBD-containing drug.

Epidiolex^®^ is an oral formulation with a recommended starting dose of 2.5 mg/kg twice daily (5 mg/kg/day), with the potential to double it after one week (up to 10 mg/kg/day), with maximum doses of 10 mg/kg twice daily (20 mg/kg/day) [33].

Dosing recommendations vary based on studies using oral administration, influenced by the specific disorder and potential interactions with other substances, like alcohol, central nervous system depressants (e.g., benzodiazepines), or moderate and potent inhibitors of CYP3A4 or CYP2C19 (e.g., clobazam and valproate) [57]. For conditions like Lennox–Gastaut syndrome [58], Dravet syndrome [59], and tuberous sclerosis complex [60], oral doses from less than 1 up to 50 mg/kg/day can be effective and are safe, even 40 mg/kg/day is well tolerated in pediatric patients with treatment-resistant epilepsy [61]. Good tolerability of CBD is evidenced in other conditions, including schizophrenia, treated with doses up to 1000 mg/day [62], nicotine addiction with 400 mg/daily [63], cannabis use disorders with 400–800 mg/daily [64], and post-traumatic stress disorder with a dose of 25–100 mg/daily [46,65]. A minimum oral dose (0.5 mg/kg/day) of CBD improved anxiety [66], and oral doses ≤ 400 mg/day ameliorated opiate addiction-related disorders [67]. Topical CBD cream positively affected skin conditions like psoriasis and acne [68], while sublingual oil drops improved the quality of life in those experiencing adverse reactions to the HPV vaccine [69]. In preclinical animal models [52], CBD exhibited efficacy in treating generalized brainstem and limbic seizures in Genetically Epilepsy-Prone Rats (GEPR) 3 strain [70] and reducing pericardial effusion and pericardial thickness in a mouse model of acute pericarditis through daily intraperitoneal injections of 10 mg/kg [71]. Nevertheless, it is noteworthy that experimentally obtained preclinical results require further validation and approval before being used in Phase 1 clinical trials.

Regarding overdosage, CBD treatment at oral doses of 15 to 160 mg, inhaled doses of 0.15 mg/kg, or injected intravenous doses of 5 to 30 mg proved to be free from side effects. In a case report, a dose equal to 1500 mg/day did not cause adverse effects in humans [52].

## 4. CBD as a Pharmacological Inhibitor

### 4.1. CBD Inhibition of Signaling in the Endocannabinoid System

CBD modifies endocannabinoid’s tone by inhibiting its signaling in a dose-dependent manner [26], demonstrating it to be a non-competitive negative allosteric modulator despite its low affinity to CB1 and CB2 receptors [17,24,26] (Figure 3). Diverse studies have explored CBD’s interactions with other proteins within the ‘endocannabinoid signaling system’ in addition to its receptors, revealing inhibitory effects on N-arachidonylethanolamide (Anandamide), the first endocannabidiol to be discovered, which is a neuromodulator with anti-inflammatory properties [17], hydrolysis by Fatty Acid Amide Hydrolase (FAAH) [72], and on Anandamide uptake through Anandamide Membrane Transporter (AMT) [73], reinforcing its inhibitory effects on inflammation [74] and favoring desensitization on sensitive neurons.

### 4.2. CBD Modulates Intracellular Ion Concentrations and Inhibitor of Ion Channels

Calcium and calcium channels

Calcium regulates a wide range of biological functions, including enzymatic activity, ion regulation, cell secretion, mitosis, and apoptosis, coordinates muscle contraction through neurotransmitter release [75], and serves as a versatile secondary messenger in various intracellular signaling pathways to maintain homeostasis in both excitable and non-excitable cells [75]. Voltage-gated calcium channels modulate neuronal excitability and promote the release of synaptic vesicles at primary afferent synapses, presenting a potential therapeutic target for addressing chronic pain [80].

CBD and other cannabinoids modulate the ryanodine-sensitive intracellular Ca^2+^ stores in neurons [9,81]. CBD directly inhibits low voltage-activated T-type calcium channels, encoded by the Ca_V_3 gene family, modifying cell excitability, mainly in nociceptive processing neurons [82]. Mice lacking the Ca_v_3.2 gene treated with intrathecally administered CBD showed no significant analgesic effect after thermal harm [80].

Additionally, CBD profoundly inhibits ATP production, lowers the threshold for Ca^2+^-induced mitochondrial transition pore activation, and inhibits mitochondrial Ca^2+^ uptake, potentially inducing cellular injury and apoptotic neurodegeneration [83]. In instances of extremely high concentrations, CBD causes mitochondrial Ca^2+^ overload, ultimately resulting in cell death [84]. Moreover, CBD elevates cytosolic-free Ca^2+^ in various cancer and non-cancerous cells, defining the scenarios related to cellular outcomes and encompassing cell fate determination, cell survival, and death [84].

In the heart, CBD depresses myocyte contractility by suppressing L-type Ca^2+^ channels at a site different than the dihydropyridine binding site and inhibits excitation-contraction coupling among cardiomyocytes [76]. CBD can limit the entry of Ca^2+^ into the mitochondria during ischemia/reperfusion by avoiding IP_3_-dependent Ca^2+^ liberation from the sarcoplasmic reticulum, preventing further injury caused by the overflow of Ca^2+^ [77,85] (Figure 3).

Sodium and sodium channels

Voltage-gated sodium channels (NaVs) allow the influx of Na+ ions and are responsible for the action potential rapid upstroke in excitable cells, most important in the central nervous system excitability and cardiomyocyte contraction [78].

CBD is a nonselective sodium-channel inhibitor of NaV1.1, NaV1.2, NaV1.6, and NaV1.7 in the low micromolar range, creating a steep average hill slope, suggesting a non-direct binding interaction [78]. CBD particularly modulates NaV1.1 currents, relevant to the epilepsy mechanism, by reducing patient seizure frequency [78].

CBD prevents the activation of sodium channels by shifting the voltage dependence of activation, increasing the depolarization potential of cells [85,86]. CBD can inhibit approximately 90% of sodium conductance while the available sodium channels remain unchanged, preventing channels from conducting without altering the activation voltage [87].

Potassium and potassium channels

CBD inhibits the inward sodium current without changing the heart action potential’s threshold, peak, and reversal potentials [86] (Figure 3).

During phase 3 of the action potential, the cardiac delayed rectifier current (IK) relies on both KV7.1, underlying the slow component (IKs), and KV11.1 (hERG), underlying the rapid component (IKr) channels, to facilitate the repolarization of the cardiomyocyte membrane, restoring its resting potential [79,86]. CBD suppresses at all potentials, both IKs and IKr, in a non-voltage-dependent manner [86,88]. Moreover, CBD exhibits no discernible impact on the inward rectifying Kir2.1 channel, which lacks voltage-sensing domains [86].

Conversely, transient outward current Ito and inward rectifier Ik1 are considerably less sensible to CBD [86,89], showing no changes in the amplitudes and kinetics of Ito and little effect on Ik1 [88,90]. This observation indirectly supports earlier study findings, which demonstrated an extended duration of action potentials at relatively low CBD concentrations and no alteration in duration at higher CBD concentrations [79] (Figure 3).

Chloride and chloride channels

CBD inhibits a recently identified mitochondrial chloride channel conductance at a low micromolar range concentration in rodents’ brain mitochondria and decreases the threshold for mitochondrial permeability transition. Therefore, it is plausible that CBD induces mitochondrial transition pore through the blockade of chloride channels, leading to the observed reduction in oxygen consumption rate and ATP production, followed by cellular death [83].

## 5. Anti-Inflammatory Effects of CBD

CBD modulates immune system cells, showcasing anti-inflammatory and antioxidant effects, closely linked to its impact on both the interferon beta (IFN-β) and the NF-κB pathway (Figure 4).

IFN-β engages the type I interferon receptor through autocrine secretion and activates signal transducers like Janus kinase (JAK) and activators of transcription (STAT)-dependent pathways [91], triggering a subsequent wave of gene expression, predominantly chemokines such as Interferon-γ inducible protein 10 kDa (CXCL10), C-C motif ligand 5 (CCL5), and C-C motif ligand 2 (CCL2).

CBD inhibits early stages of IFN-β synthesis and stimulates interferon-regulating factor 3 (IRF-3), which involves phosphorylation by TANK-binding kinase 1 (TBK1) facilitated by the TIR-domain-containing adapter-inducing interferon-β (TRIF) adaptor protein associated with TLR4 receptors. Activated IRF-3 binds to the IFN-stimulated response elements–DNA sequence, culminating in the production of the IFN-β cytokine [91]. Additionally, CBD enhances inhibitory STAT3 phosphorylation while concurrently reducing pro-inflammatory STAT1 phosphorylation.

The NF-κB pathway, crucial in cellular signaling, involves inducible transcription factors NF-κB1 (p50), NF-κB2 (p52), RelA (p65), RelB, and c-Rel [92], forming various hetero- or homodimers and binding to the κB enhancer on DNA. Under normal conditions, NF-κB proteins are sequestered in the cytoplasm by inhibitory proteins like the IκB family [92]. NF-κB proteins serve as a critical regulator of pro-inflammatory gene expression and act as a central transcription factor for M1 macrophages, inducing the expression of inflammatory genes such as TNF-α, IL-1β, IL-6, IL-12p40, and cyclooxygenase-2 [91]. Activation of the NF-κB pathway in response to LPS via Toll-like receptor 4 (TLR 4) involves IκB inactivation through Interleukin-1 receptor-associated kinase 1 (IRAK-1)-dependent phosphorylation, followed by ubiquitin-dependent degradation of both IRAK-1 and IκB. This leads to NF-κB p65 phosphorylation and nuclear translocation [91,93,94]. Additionally, canonical NF-κB members, RelA and c-Rel, play a significant role in mediating TCR signaling and initiating naive T-cell activation [93].

CBD attenuates IRAK-1 degradation and reverses IκB degradation, resulting in a decrease in NF-κB p65 nuclear translocation. CBD impacts the interplay between transcription factors, such as nuclear factor erythroid 2-related factor 2 (Nrf2) and NF-κB, enhancing the expression of Nrf2 activators and promoting its transcriptional activity [5], lastly, inhibiting the NF-κB pathway.

CBD may mitigate inflammatory responses through a multifaceted modulation of key components within the NF-κB signaling network and interactions with other relevant transcription factors. The collective impact on these key immunological pathways highlights the potential of CBD as a versatile compound with implications for therapeutic interventions in inflammatory processes.

### Additional Molecular and Cellular Effects of CBD

Nuclear peroxisome proliferator-activated receptors (PPAR)

PPARγ, a key regulator of cell metabolism, modulates mitochondrial function and the inflammatory response. It inhibits pro-inflammatory cytokines [95,96] while elevating anti-inflammatory cytokines, suppresses inducible nitric oxide synthase (iNOS) expression from macrophages [97,98,99], dendritic cells [100,101], T cells [95,102], and B cells [95]. PPARγ NF-κB-inhibiting IκB kinase [95,103] also regulates lipid and glucose homeostasis, serving as a pivotal transcription factor in adipocyte differentiation [104]. PPARγ is highly expressed in activated peritoneal macrophages and macrophage-derived foam cells within human atherosclerotic lesions [97,98].

CBD is a functional PPARγ agonist, increasing its transcriptional activity [105]. Through PPARγ, CBD reduces endothelial cell activation and monocyte adhesion by reducing the vascular cell adhesion molecule (VCAM) expression and trans-endothelial migration [35,106,107,108,109]. Also, it decreases PPARγ-mediated NFκB activation [110,111]. CBD increases lipid accumulation and pro-adipogenic genes in mice and humans, inducing adipogenic differentiation through a PPARγ-dependent mechanism [112]. PPARγ is involved in CBD’s proapoptotic and tumor-regressive action. CBD prompts increased COX-2–dependent prostaglandin levels in lung tumor cell lines, leading to PPARγ translocation to the nucleus and inducing PPARγ-dependent apoptotic cell death [108].

The evidence suggests that PPARγ is a critical factor in CBD’s ability to modify inflammatory responses, oxygen damage, adipose tissue homeostasis, and cell survival, among others.

Adenosine receptors

During cellular stress and inflammation, adenosine acts as an endogenous regulator, triggering an autoregulatory loop wherein immunosuppression safeguards organs from injury caused by the immune response [113]. In the cardiovascular system, adenosine is a potent negative inotropic agent and a coronary vasodilator [113] and regulates T-cell proliferation and cytokine production [114].

Adenosine transporters are sorted into two categories: equilibrative nucleoside transporters (ENTs) and concentrative nucleoside transporters (CNTs). ENTs, particularly ENT1, are highly expressed in the central nervous system, while CNTs, namely the A1, A2A, A2B, and A3 adenosine receptors, have been identified in various cells and tissues [114].

CBD acts as a competitive inhibitor at the ENT1 transporter, affecting adenosine uptake-enhancing endogenous adenosinergic signaling [115], as evidenced in macrophages, neurons, microglial cells, and the myocardium [115,116,117]. CBD also acts through adenosine receptors A1 and A2A [88,116] and modulates its activation [89,118,119,120].

GABA^A^ receptors

Gamma-aminobutyric acid (GABA) is a neurotransmitter highly concentrated in animal brains [121]. In mature neurons, GABA acts by activating chloride (Cl^−^)-permeable GABA_A_ receptor channels, leading to membrane hyperpolarization [122]. GABAA receptors’ distribution in the brain is regionally specific, with different subunit combinations mediating distinct roles. GABA impairment is a well-established mechanism linked to brain hyperexcitability, notably observed in epileptic disorders [123].

CBD is a positive allosteric modulator, amplifying currents induced by low GABA concentrations more than those induced by higher concentrations [121]. CBD significantly and reversibly amplifies GABAergic-evoked currents in most frequent refractory epileptic diseases, suggesting potential subtype or receptor arrangement differences [123].

## 6. CBD as a Pharmacological Inhibitor, an Overview of Preclinical and Clinical Studies

### 6.1. CBD in Neurodegenerative and Psychiatric Disorders

Epilepsy

Epilepsy, a neurological disorder characterized by sudden, recurrent, and unprovoked seizures, has been the focus of CBD research since the 1970s [124,125], showing promising outcomes. Four recent clinical trials employing pharmaceutical-grade, plant-derived CBD (Epidiolex^®^) led to its approval for treating treatment-resistant seizures in neurodegenerative syndromes like Dravet Syndrome, Lennox–Gastaut Syndrome, and Tuberous Sclerosis complex [55,58,59,126,127]. These clinical trials showed CBD’s efficacy, safety, and tolerability over extended periods [60,128,129].

The diverse molecular targets of CBD make it challenging to pinpoint the exact mechanisms behind its anti-epileptic effects. Animal models suggest CBD’s involvement in GPR55 antagonism [130], TRPV1 desensitization [131], inhibition of adenosine reuptake [132], reduction in NMDA-induced convulsive syndrome through the sigma 1 receptor (σ1R) [133], and enhanced GABAA receptor activation [134]. CBD also increased resistance to induced seizures in a mouse model of epilepsy associated with a mutation in the *Scn8A gene*, encoding Na_v_1.6, a subtype of Na_v_s [128] (Figure 5).

Alzheimer’s Disease

The brain of Alzheimer’s disease (AD) patients is characterized by Amyloid β (Aβ) accumulation and hyperphosphorylation of tau protein, causing neuroinflammation and oxidative stress leading to neurodegeneration and cognitive impairment [135]. In animal models of AD with Aβ injections, the effect of CBD proves to be anti-inflammatory, acting through the reduction in mRNA expression of glial fibrillary acidic protein (GFAP), NO, and release of IL-1β [136], and through the activity on the PPAR-γ receptor [110]. Furthermore, CBD treatment reversed the cognitive deficits of Aβ-treated and AD-transgenic mice [137,138,139]. Due to the preclinical evidence, current clinical trials are testing the effects of CBD on behavioral symptoms in patients with Alzheimer’s dementia [140,141] (Figure 5).

Parkinson’s Disease

Parkinson’s disease (PD), a neurodegenerative disorder marked by motor system impairment, progressive dementia, and depression, has seen potential neuroprotective benefits in animal models from CBD’s antioxidant properties [142]. In vivo experiments demonstrated CBD’s positive impact on behavioral impairment and anti-cataleptic effects, possibly through 5-HT1A activation [143,144]. CBD’s influence on TRPV1 might contribute to reducing neuroinflammation and enhancing the motor system [145]. Given its preclinical outcomes, some clinical trials have explored CBD’s tolerability and efficacy in alleviating motor symptoms and tremors in PD patients [146,147] (Figure 5).

Huntington’s Disease

Huntington’s disease (HD) is a genetic neurodegenerative disorder characterized by motor and behavioral impairment. In an animal model of HD, the anti-oxidant properties of CBD promoted neuroprotection against striatal degeneration caused by the injection of a mitochondrial complex II inhibitor (3-nitropropionic acid) [148]. In another model of HD induced by malonate treatment, only CB2 receptor agonists were able to show neuroprotection [149]. However, in clinical trials, CBD was more effective when administered with Δ9-THC [150] than alone [151] (Figure 5).

Schizophrenia

Schizophrenia (SCZ), a heterogeneous disorder emerging in late adolescence or early adulthood, has seen positive outcomes in CBD administration studies for improving positive psychotic symptoms with low side effects [152,153]. Experimental animal studies of CBD in SCZ models demonstrated reduced hyper-locomotor activity induced by dopamine receptor agonists and alleviated catalepsy [154,155,156]. CBD’s effects on pre-pulse inhibition (PPI) [157,158], measuring sensorimotor gating modifications associated with SCZ symptoms [152,153], showed attenuation in various animal models, including those induced by amphetamine [157,158] or MK-801, a non-competitive agonist of the NMDA receptor [155,159,160]. However, in neuregulin 1 (NRG1) mutant mice, a genetic SCZ model, CBD increased social interactions but showed no effects on PPI alterations [153] (Figure 5).

Anxiety and depression

CBD’s anti-anxiolytic and anti-depressant effects have been extensively demonstrated in various preclinical models and clinical trials, showing controversial results influenced mainly by factors like dose, administration time, animal models, strain, gender, and age [152]. However, the central common mechanisms of action found were the regulation of the 5-HT1A receptor [144,161,162,163,164,165] and endocannabinoid system signaling [166,167]. Clinical trial results are limited, but ongoing trials are exploring the anti-anxiety and anti-depressant effects of CBD [152,153] (Figure 5).

### 6.2. CBD in the Treatment of Pain and Autoinflammatory Diseases

As previously described, CBD exerts anti-inflammatory properties at molecular and cellular levels. For this reason, CBD has been proposed for the treatment of symptoms in several autoimmune diseases, including arthritis, Inflammatory Bowel Disease, and multiple sclerosis.

Arthritis

Animal studies on osteoarthritis primarily focused on CBD’s analgesic properties for pain control [168,169,170,171]. CBD has been explored as a potential treatment for Rheumatoid arthritis, demonstrating anti-inflammatory and analgesic effects in collagen-induced arthritis animal models [172], increasing intracellular calcium levels, and decreasing cell viability and pro-inflammatory cytokines production [173]. Clinical trials on CBD for arthritis are limited, with ongoing studies [174,175]. In a cross-sectional study including patients with different types of arthritis, CBD showed benefits in pain relief, physical function, and sleep quality, particularly in the osteoarthritis group [176].

However, a different trial failed to significantly improve quality of life and pain management in patients with hand osteoarthritis and psoriatic arthritis [177] (Figure 5).


*Inflammatory Bowel Diseases*


Inflammatory Bowel Diseases (IBD), including Ulcerative Colitis (UC) and Crohn’s Disease (CD), involve chronic inflammation of the gastrointestinal (GI) tract. CBD, acting on CB1 and CB2 receptors highly expressed in the GI tract [178], shows potential in IBD management [179], since CBD, via PPAR-γ, reduces iNOS [180] and S-100 protein expression [181], myeloperoxidase activity [182], and increased cytokine production (IL-1β, IL-10) [180,183], preventing gut permeability due to inflammation [184] and morphological changes [185], decreasing colon damage and intestinal hypermotility [182,183,186]. Despite the beneficial effects shown in these experimental models, human trials improved quality of life with no significant changes in inflammation or endoscopic scores [187,188,189] (Figure 5).


*Multiple sclerosis*


Multiple sclerosis (MS) is an autoimmune neurodegenerative disease of the central nervous system (CNS). In the Experimental Autoimmune Encephalomyelitis (EAE) murine model, CBD improved clinical signs of MS, suppressing the microglial activity and T-cell proliferation [190]. In this model, CBD reduced neuronal apoptosis by inhibiting Fas activation, extracellular signal-regulated kinase (ERK) phosphorylation, and caspase-3 activation, modulating the Bax/Bcl2 ratio and reducing p53-p21 activation [191]. Additionally, CBD modulated the Phosphoinositide 3-kinases (PI3K)/Protein kinase B (Akt)/Mammalian target of rapamycin (mTOR) pathway, reduced IL-17 and INF-γ, up-regulated PPAR-γ, inhibited the transcription factor c-JUN and p38 MAP kinase [192], and increased myeloid-derived suppressor cells (MDSCs) [193,194,195]. In a Theiler’s murine encephalomyelitis virus murine model, CBD given at the onset of infection ameliorated motor deficit, reduced microglial activation and production of IL-1β, and infiltration of leukocytes in the CNS by modulating VCAM-1 and CCL2 and CCL5 expression, effects partially reversed by an Adenosine receptor antagonist [120]. Most clinical studies in MS are exploring the effects of CBD in combination with THC [196], whereas only one clinical study is recruiting MS patients to study the effects of CBD on sleep and pain [197] (Figure 5).

### 6.3. CBD in Heart Disease

Ischemic heart disease

Ischemic heart disease is the leading cause of heart failure and cardiovascular morbidity. CBD protects against myocardial ischemia-reperfusion (I/R) injury in different animal models in vivo and ex vivo [198,199,200,201,202]. CBD reduces infarct size, improves post-myocardial infarction (MI) left ventricular function, and attenuates inflammatory response [202,203]. CBD also improves microvascular perfusion and reduces microvascular obstruction [116]. The latter effect may be due to the ability of CBD to target platelets. In fact, it reduces platelet activation and aggregation of human and rabbit platelets in vitro and rat platelets before an ischemic event [204] (Figure 5).

Toxic cardiomyopathy induced by Doxorubicin

CBD protects from doxorubicin-induced cardiomyopathy in rats and mice [205,206]. This anti-cancer antibiotic induces free radical generation and apoptosis damages mitochondria and cardiomyocytes, and promotes an inflammatory response mediated by NF-κB and the NLRP3 inflammasome pathway [207,208]. CBD reduces all these effects of doxorubicin [206] (Figure 5).

Myocarditis and pericarditis

In a mouse myocarditis model, CBD reduces inflammatory cell myocardial infiltration and the expression of pro-inflammatory cytokines, ventricular dysfunction, and fibrosis [209,210]. In a pericarditis mouse model, CBD reduced pericardial effusion and thickness [71]. A pharmaceutically manufactured CBD formulation (CardiolRx™, Oakville, ON, Canada) is undergoing clinical testing for the treatment of acute myocarditis and pericarditis [211,212] (Figure 5).

Arrhythmias

Despite the consistent cardioprotective effects reported for CBD, there is a concern that it may increase arrhythmic risk in patients. Although this risk has not been determined in humans, CBD could induce direct electrophysiological responses affecting calcium, sodium, and potassium channels [87]. It also delayed IKs in ventricular myocytes in vitro, prolonging cardiac action potential and QT interval, possibly leading to an arrhythmogenic effect [204]. CBD increases the heart rate in healthy subjects [213,214]. On the contrary, CBD decreased the incidence of ectopic beats and ventricular fibrillation in animal models of myocardial I/R injury [202] (Figure 5).

Vasodilatory effects and Hypertension

In animal models, CBD induces vasodilatation [105,215], and in humans, it induces relaxation of pre-constricted mesenteric arteries [216]. While studies with anesthetized animals showed no significant impact on basal median arterial pressure or heart rate [217,218], CBD reduced artery pressure and heart rate in restraint animals [219]. Following myocardial I/R injury, CBD decreases arterial blood pressure [204]. No significant effects on blood pressure and heart rate were observed in human trials where blood pressure changes were not a significant outcome measured [220,221,222]. However, in a trial designed to measure the hemodynamic effects of CBD in healthy volunteers, CBD reduced resting systolic and diastolic blood pressure [213]. Similarly, CBD reduced the systolic and diastolic blood pressures in patients with mild or moderate hypertension [223] (Figure 5).

### 6.4. Diabetes

Diabetes, classified into type 1 (caused by loss of pancreatic beta cells) and type 2 (caused by insulin resistance secondary to metabolic disorders), is always characterized by a hyperglycemic state that affects several organs and systems [224].

Diabetic cardiomyopathy

Diabetic cardiomyopathy involves structural and functional pathologic changes in the heart independently of classical cardiac risk factors, such as hypertension, hyperlipidemia, coronary artery, or valvular diseases [225]. Hyperglycemia promotes vascular damage, endothelial dysfunction, oxidative damage, leukocyte homing, and inflammation [225,226,227]. CBD attenuates these changes and reduces myocardial fibrosis in models of diabetic cardiomyopathy model [226,228,229]. CBD preserves myocardial systolic and diastolic function in mice with type 1 diabetes. CBD also reduces the incidence of diabetes and neurotoxicity and preserves the blood-retinal barrier in mice [228,230]. However, in human subjects, CBD did not affect glycemic control or insulin sensitivity [228] (Figure 5).

Diabetic nephropathy

The onset of diabetic nephropathy is a significant cause of renal failure. Again, oxidative stress, inflammation, and fibrosis are major factors affecting renal function during diabetes. CBD reduces kidney leukocyte infiltration without affecting kidney fibrosis [231]. It has been observed that CBD worsens kidney function in mice with type 1 diabetes [231]. However, in human trials, CBD did not promote adverse kidney effects [188,232] (Figure 5).

### 6.5. CBD in COVID-19

CBD exhibits anti-inflammatory properties, reducing lung inflammation in animal models of LPS-induced acute lung injury by the reduction of leukocyte migration, myeloperoxidase activity, and release of inflammatory cytokines (TNF and IL-6), as well as chemokines (MCP-1 and MIP-2) [233], possible through the Adenosine A2_A_ receptor [119]. SARS-CoV-2 is an RNA virus infecting human cells by binding the spike (S) protein to the angiotensin-converting enzyme 2 (ACE2) receptor. In vitro experiments, CBD exhibited anti-viral properties, inhibiting viral replication [234,235] and decreasing levels of ACE2 protein [236]. The cytokine storm driven by COVID-19 can cause acute respiratory distress syndrome (ARDS), leading to organ failure. CBD reduces the production of pro-inflammatory cytokines and protects lung tissues in animal models infected with a synthetic double-stranded RNA, Poly I: C-induced ARDS [237,238]. Despite the promising preclinical results, in a clinical trial where 105 patients were randomly allocated in the CBD (300 mg/day for 14 days) or placebo group, CBD was well tolerated; still, no significant differences were found in the severity of disease or median time of symptoms resolution between the two groups [239]. Other clinical trials are currently ongoing to explore the potential of CBD on pulmonary infection [237], cardiovascular diseases and risk factors [238], and treatment of long COVID [240] (Figure 5).

**Figure 5 molecules-29-00473-f005:**
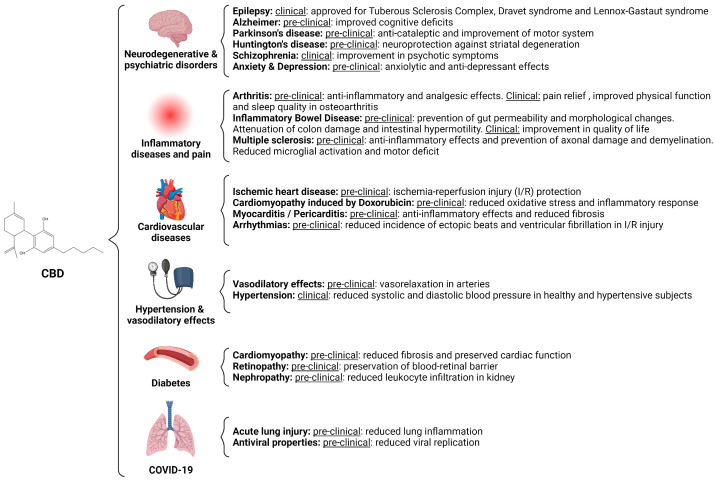
Summary of the effects of CBD in different organ systems and diseases such as neurodegenerative conditions, pain management, cerebral ischemia-reperfusion injury, heart diseases, vascular diseases, diabetes, and hypertension [54,57,58,71,104,110,125,126,137,138,144,148,151,152,167,168,169,170,181,182,185,201,202,205,211,214,215,218,219,220,221,224,226,227,229,232,233,235,236,241].

## 7. Conclusions

The pharmacokinetics of CBD reveal a complex interplay of factors, like various formulations, routes of administration, dosage schedules, and individual characteristics. Despite its widespread use, oral administration poses challenges due to limited bioavailability, which has prompted the development of advanced formulations yielding greater absorption. Inhalation offers rapid absorption with higher bioavailability, while intravenous administration produces the highest plasma concentrations. Transdermal administration, although challenging due to CBD’s hydrophobic nature, has shown promise with enhanced permeation strategies. The distribution of CBD, being highly lipophilic, leads to its sequestration in fatty tissues and penetration into various organs, providing the largest accumulation and possibly long-lasting effects. However, its safety profile is generally favorable.

CBD’s role as a pharmacological inhibitor extends to its modulation of the endocannabinoid system and ion channels, including calcium, sodium, potassium, and chloride channels. Anti-inflammatory and pro-survival activities of CBD are due to its ability to inhibit several pro-inflammatory and cell-death pathways, such as interferon beta and NF-κB, showcasing its potential as a versatile compound in immune system regulation and therapeutic interventions for inflammatory processes. Moreover, it has been shown that CBD is a potent activator of several membrane and intracellular receptors with inhibitory functions.

CBD has demonstrated a wide range of potential therapeutic effects in both preclinical and clinical studies across various neurological, psychiatric, autoimmune, and cardiovascular disorders. The pharmacological inhibitory properties of CBD, combined with its anti-inflammatory, antiapoptotic, and antioxidant characteristics, make it a versatile compound with diverse applications. Despite promising results, further research, especially large-scale clinical trials, is necessary to establish CBD’s efficacy and optimal usage for specific medical conditions.

As CBD continues to garner attention, ongoing studies will likely provide additional insights into its mechanisms of action, allowing for more precise and effective therapeutic applications.

## Figures and Tables

**Figure 1 molecules-29-00473-f001:**
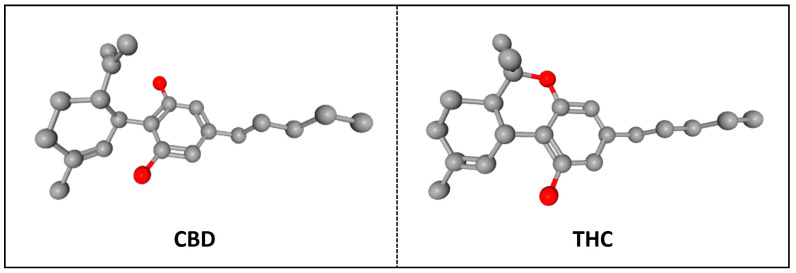
3D chemical structures of CBD (PubChem CID: 644019 [29]) and Δ9-THC (PubChem CID: 16078 [30]). The main difference between both compounds is that CBD contains a hydroxyl group while THC possesses a cyclic ring. Gray spheres represent carbon atoms, and red spheres represent oxygen atoms [9,31].

**Figure 2 molecules-29-00473-f002:**
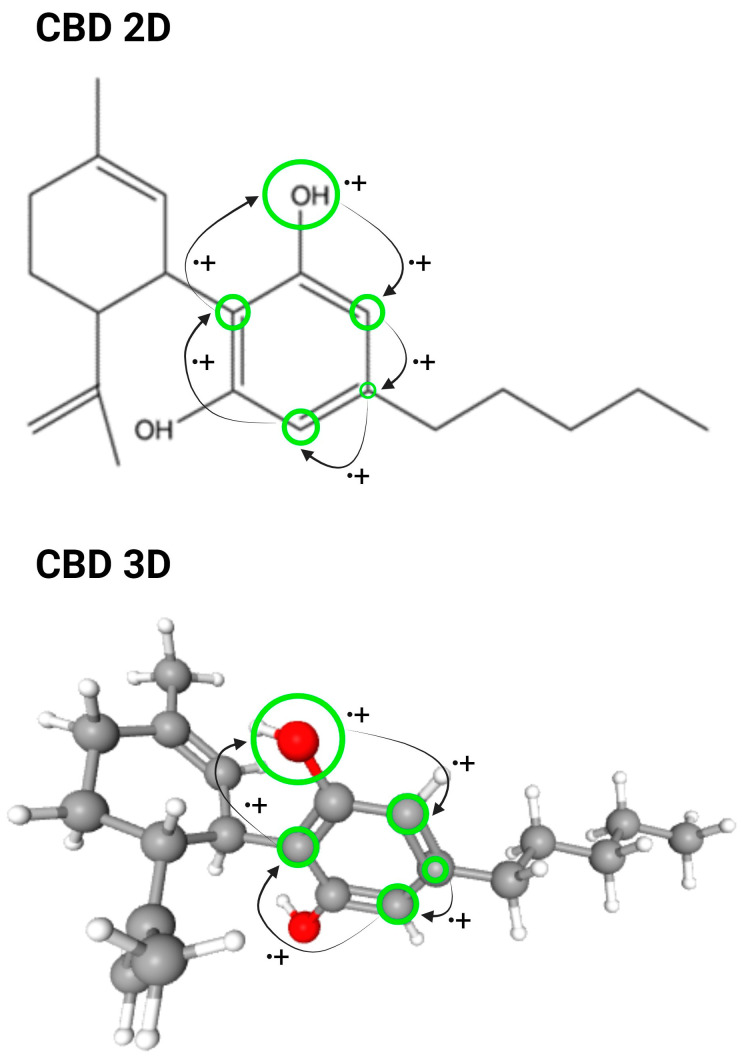
CBD structure. Carbons atoms are represented in grey, Oxygen atoms in red and Hydrogen atoms in white. The arrangement of the cation-free radicals in CBD is illustrated through green circles, and the arrows indicate the resonance of electrons within the molecule in both 2D and 3D images. These sites on ether and alkyl moieties and in the benzene ring are potentially responsible for the antioxidant properties of CBD [27,28,32].

**Figure 3 molecules-29-00473-f003:**
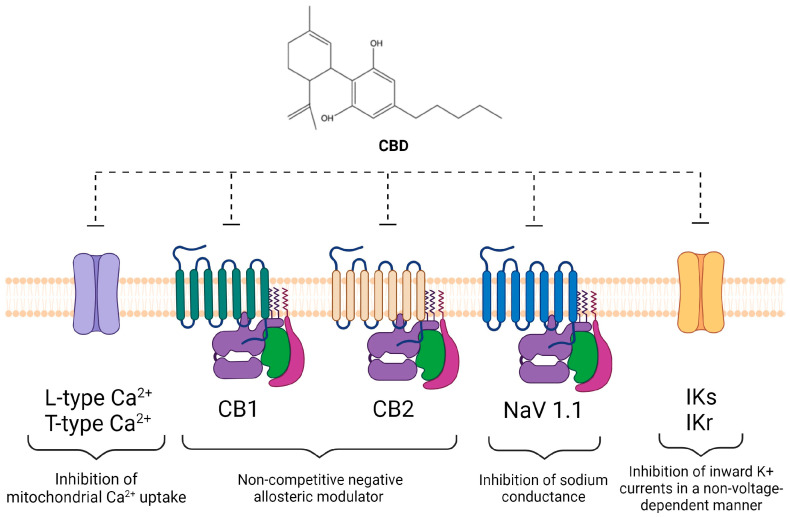
Main targets of CBD inhibitory function. Schematic figure with the ion channels and the receptors inhibited by CBD with reported intracellular effects [16,23,25,74,75,76,77,78,79].

**Figure 4 molecules-29-00473-f004:**
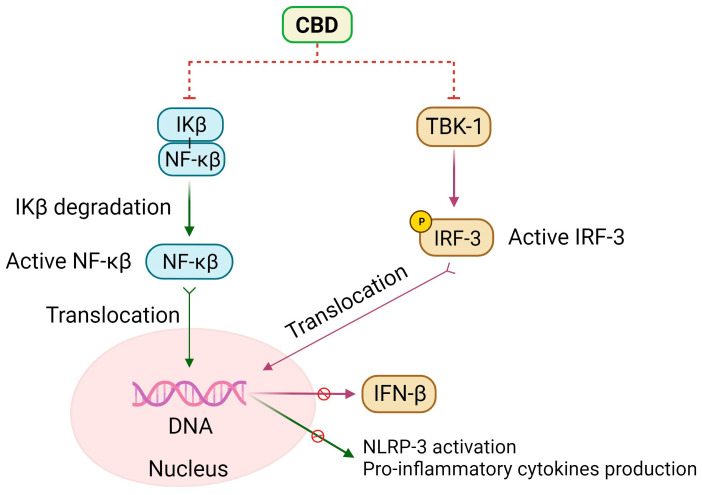
CBD inhibits NF-kB and IFN-β signaling. CBD prevents IκB degradation, reducing NF-kβ activation and translocation, which blocks the translation of several inflammatory proteins, including the NLRP-3 inflammasome priming and the consequent pro-inflammatory cytokine production. CBD also inhibits TBK-1 activation and IRF-3 phosphorylation and activation. Finally, as IRF-3 translocation to the nucleus is inhibited, the expression of IFN-β is reduced [4,90,91,92,93].

## Data Availability

No datasets were generated or analyzed during the current study.

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
