# Peer review of "An Overview of Cannabidiol as a Multifunctional Drug: Pharmacokinetics and Cellular Effects"

_molecules, 2024, doi:10.3390/molecules29020473_

Round 1
Reviewer 1 Report
Comments and Suggestions for Authors
This review is of great interest and describes in detail cannabidiol, a non-psychoactive compound derived from Cannabis Sativa. The authors describe CBD's structure, its pharmacokinetics, its role as a pharmacological inhibitor, the molecular mechanisms on which it acts, and its anti-inflammatory effects. In this study, different routes of administration of CBD are and compared.
The authors highlighted a wide range of potential therapeutic effects that CBD has demonstrated in preclinical and clinical studies on various neurological, psychiatric, autoimmune and cardiovascular disorders.
1) Introduction
This paragraph is clear, the cited references are mostly recent publications and relevant.
2) CBD: structure, sources, and clinical use.
This paragraph is well written and clarifies precisely, from a chemical point of view, the antioxidant properties of CBD. This chemical description is useful to better understand the function of CBD.
3) CBD as a drug: pharmacokinetics and current formulations
This section discusses the routes of administration, dosing schedule (single or multiple doses), type of formulation system and dietary factors. The authors point out that all these factors play a significant role in influencing the pharmacokinetics and effects associated with cannabis-derived drugs. This part analyses the pharmacokinetics of CBD, adding to the information in the literature.
4) CBD as a Pharmacological Inhibitor
In this section CBD's role as a pharmacological inhibitor is explored, encompassing interactions with the endocannabinoid system and ion channels. This paragraph is well written and useful to clary the role of CBD.
5) Anti-Inflammatory Effects of CBD
In this section, the authors described clearly, CBD's anti-inflammatory and immune-regulating effects on interferon-beta and NF-κB. The molecular mechanisms of activation of the interferon beta (IFN-β) and the NF-κB pathway are well explained.
6) CBD as a Pharmacological Inhibitor, An Overview of Preclinical and Clinical Studies
The authors present a well-written summary of the effects of CBD in various organ systems and diseases such as neurodegenerative conditions, pain management, cerebral ischaemic-reperfusion injury, heart disease vascular diseases, diabetes and hypertension.
7) Conclusions
The statements and conclusions drawn coherent and supported by the listed citations and prospects are well described.
This review is relevant and of great interest to the scientific community; moreover it is clear, comprehensive and of relevance in the field of cannabinoids.
The strength of this review lies in the fact that it studies critically several aspects of CBD: its structure, its pharmacokinetics, its role as a pharmacological inhibitor, the molecular mechanisms on which it acts and its anti-inflammatory effects. For this reason, this review contributes to increasing knowledge on cannabinoids and their mechanisms of action.
The cited references are mostly recent publications and relevant. The figures are appropriate, easy to interpret and understand.
The author presents a detailed manuscript, that is of high quality, in particular the topic discussed is of great interest, original and I recommend its publication on molecules.
Author Response
Dear reviewer,
Thank you for taking the time to review our manuscript thoroughly. We are sincerely grateful for your thoughtful comments and constructive feedback. Your positive evaluation is immensely encouraging, and we appreciate your recognition of our work's relevance and interest in the scientific community.
We are honored by your recommendation to publish our manuscript on molecules.
Reviewer 2 Report
Comments and Suggestions for Authors
To the authors, I continue with the following considerations:
- check the writing of species names, they must be highlighted, and the second name, in lowercase,
- standardize references
- at the beginning of the introduction, they use the same references (1-3) in two paragraphs, due to the theme of the sentence, I suggest combining the two or only citing them the second time, so as not to be repetitive,
- regarding the use of CBD, I recommend that although several studies attest to its safety, be careful when citing doses, and highlight that the results obtained experimentally require further validation before considering clinical use (to indicate in the text),
- include which studies the statements in Figure 4 were based on, in the caption, even for the other figures
- the work addresses several therapeutic possibilities regarding CBD, it is suggested to insert a paragraph in the introduction about a brief mention of the historical interest in Cannabis compounds, taking into account the different discussions about the benefits and harms, considering that several people ancients use cannabis, not for isolated use of CBD, but seeking the same therapeutic interests now associated with this compound
Author Response
Dear Reviewer 2,
We appreciate your valuable comments and have diligently incorporated the suggested changes to enhance the clarity and completeness of our manuscript.
Here is a summary of the modifications made in response to your feedback:
- We revised the presentation of species' Latin scientific names, now highlighting both the capitalized genus name and the non-capitalized specific epithet in italics throughout the text.
- The references in the text underwent a thorough check, correction, and standardization process to ensure accuracy and consistency.
- We included explicit references to the studies on which the figures are based, providing a more transparent linkage between the visuals and their underlying research.
- In response to your specific recommendation regarding the use of CBD, we added a clarifying paragraph on page 6 (lines 241-242) to address the need for caution when citing doses.
- Regarding the suggestion to include a brief historical context in the introduction about the evolving interest in Cannabis compounds, we added a paragraph highlighting the temporal evolution of Cannabis use, considering the historical discussions surrounding its benefits and harms (see page 1, lines 34-40).
We believe these modifications have significantly improved our manuscript's overall content and quality. We hope these adjustments align with your expectations and address any concerns raised during the review process. We are committed to ensuring that our work meets the standards set by the editor and the reviewers.
Reviewer 3 Report
Comments and Suggestions for Authors
Overall, this regular manuscript by Martinez-Naya et al. is well written. There are sufficient data which is reflected for publication in the “Journal of Molecular Structure” in suitably. So, I recommend to Accept this manuscript as current form.
Author Response
Dear Reviewer 3,
Thank you for your positive evaluation and recommendation for the acceptance of the manuscript. We sincerely appreciate your time and consideration.
Reviewer 4 Report
Comments and Suggestions for Authors
Dear Authors!
I read your manuscript very carefully and found it very interesting. The various effects of CBD could be promising for regulating different disorders. Your review paper is well-organised and -written, summarizing the existing literature about the CBD effects. I consider it suitable for publication in the current form.
Author Response
Dear Reviewer 4,
We value your insights and commendations and are grateful for the time and effort you dedicated to reviewing our work.
We are delighted to hear that you found the content, organization and writing style of our review paper suitable for this Journal.